# Using Mathematical Models to Study the Influences of Different Ratios of Chemical Nitrogen, Phosphorus, and Potassium on the Content of Soluble Protein, Vitamin C, and Soluble Sugar in Melon

**DOI:** 10.3390/ijerph20010283

**Published:** 2022-12-24

**Authors:** Shuangxi Li, Zhaohui Zhang, Juanqin Zhang, Xianqing Zheng, Hanlin Zhang, Haiyun Zhang, Yue Zhang, Naling Bai, Weiguang Lv

**Affiliations:** 1Eco-Environmental Protection Research Institute, Shanghai Academy of Agricultural Sciences, Shanghai 201403, China; 2Agricultural Environment and Farmland Conservation Experiment Station of Ministry of Agriculture and Rural Affairs, Shanghai 201403, China; 3Shanghai Key Laboratory of Protected Horticultural Technology, Shanghai 201403, China; 4Shanghai Environmental Protection Monitoring Station of Agriculture, Shanghai 201403, China; 5Key Laboratory of Low-Carbon Green Agriculture in Southeastern China, Ministry of Agriculture and Rural Affairs, Shanghai 201403, China; 6Zhuanghang Integrated Experiment Station, Shanghai Academy of Agricultural Sciences, Shanghai 201415, China

**Keywords:** balanced fertilization, melon quality, mathematic model, regression equation

## Abstract

The current fertilizer recommendations for melon plantation have many limitations and exhibit deficiencies regarding future development. Therefore, in this study, the optimal quantities of fertilizer, in terms of the effects of single factors and interaction effects, are studied. There were significant interaction effects between N and P, N and K, P and K; the contents of soluble protein, vitamin C (Vc), and soluble sugar in melon could be improved using the optimal fertilization ratios. The optimal ratio of N:P:K was 2.33:1:3.85, with the amounts of N, P_2_O_5_, and K_2_O, respectively, being 157.5, 67.58, and 260.38 kg/hm^2^, yielding 8.73 g/kg of soluble protein in melon. The optimal ratio of N:P:K was 2.03:1:3.36, with amounts of N, P_2_O_5_, and K_2_O being 157.50, 77.40, and 260.38 kg/hm^2^, respectively, yielding 25.32 g/kg Vc content in melon. Finally, the optimal ratio of N:P:K was 1.53:1:3.36, with the amounts of N, P_2_O_5_, and K_2_O being 118.07, 77.40, respectively, and 260.38 kg/hm^2^, yielding 13.34% soluble sugar content in melon.

## 1. Introduction

Irrigation and fertilization are the main stress factors that restrict the growth of crops worldwide [1]. The suitable application of fertilizers plays a vital role in sustainable agricultural production and food safety [2]. The yields of global crops would be reduced by 40–50% if the utilization of nitrogen (N) and phosphorus (P) fertilizers was suddenly suspended. The excessive and unreasonable application of chemical fertilizers, however, often decreases the quality of fruit and vegetables, as well as causing agricultural non-point-source pollution, leading to the contamination of surface water and groundwater [3]. In this aspect, precise fertilization through the use of mathematical models is highly encouraged for the cultivation of a variety of food crops [4], cash crops [5,6], and horticultural crops [7], which is expected to contribute to raising the efficiency of fertilizer utilization, reducing agricultural pollution, and improving crop quality.

*Cucumis melo* L., a common fruit crop, is an annual plant of the gourd family. The fruit is productive and popular worldwide. In particular, China is the largest melon producer in the world. Many previous studies concerning the cultivation of melons have been reported, focusing on aspects such as climatic adaptation [8], crop rotation selection, and fertilizer type and management [9,10,11]. Different mathematical models have been analyzed for melon cultivation, in order to establish a precisive pattern of fertilizer application to improve the growth, quality, and resistance to pests and diseases [6,12,13,14]. At present, in China, the fertilizer recommendations for melon cultivation are mostly based on empirical tables, results of soil nutrient analyses or field experiments. These methods, however, are highly limited by geographical restrictions, low flexibility, and high cost. In fact, the recommended quantities of fertilizers and the ratios of nutrients show no future popularization, since they often do not vary with the expected yields and soil properties.

So far, less work has considered the effects of proper fertilization on improving the quality of melon fruits. For this study, a field experiment was carried out to analyze the influences of different N, P, and potassium (K) ratios on the contents of soluble protein, vitamin C (Vc), and soluble sugar in melon fruits. We aim to establish the best model of fertilization for *Cucumis melo* L. cultivation, not only to produce high-quality fruits, but also to reduce the application of chemical fertilizers.

## 2. Material and Methods

A field experiment was carried out at the Zhuanghang Integrated Experiment Station of Shanghai Academy of Agricultural Sciences, Shanghai, China (30°53′ N, 121°23′ E), from February to November 2016. The general climate conditions of the experiment plot are as follows: the annual average temperature is 15.5 °C; the annual average sunshine hours are 1971 h; the average annual precipitation is 1047.3 mm; the annual average rainfall day is 114.8 d; and the annual average frost-free period is 229.7 d. The contents of available N (AN), available P (AP), and available K (AK) in the experimental soil were 120.60, 96.65, and 180.01 mg/kg, respectively. The melon seeds were provided by the Institute of Facilities and Horticulture, Shanghai Academy of Agricultural Sciences. The utilization rate of fertilizers was list in Table 1.

Three factors and twenty-three fertilizer treatments were designed for the experiment (Table 2). Each treatment was undertaken for three replications. As a result, 69 test plots were built in the field, with each being 1.4 m × 2.3 m in area. The plots were separated using plastic film, in order to prevent the inter-infiltration or mixture of fertilizers between each other.

Before plantation, the base fertilizers were applied into the plots. The melon seeds were sowed at the plots on 18 February 2016. After density adjustment, ten seedlings of melon were left in each plot on 22 March 2016. The growth cycle of melon is 90–100 d. According to the growth characteristics and agronomic requirements of melon, only 10 plants could be planted in each plot, and were all harvested in the maturity stage. One fruit per melon plant was left and also used for the sampling analysis. Carbamide (Weihe River, Shaanxi Province, China), calcium superphosphate (Zhejiang Jiashan Chemical Fertilizer Co., Ltd., Zhejiang, China), and potassium sulphate (Shanghai Yongtong Chemical Industry, Shanghai, China) were applied as the N, P, and K fertilizers in this experiment, respectively. The rates of fertilizers applied for different treatments are described in Table 2 and Table 3.

For each treatment, one-third of the nitrogen fertilizer (carbamide) was used as base fertilizer, and the rest was evenly divided and applied as top-dressing during the extension and enlargement stages of melon (spreading the fertilizer along a ditch). All phosphate fertilizers (calcium superphosphate) were applied as base fertilizer. Half the amount of the potassium fertilizers (potassium sulfate) was applied as base fertilizer, and the rest was applied as top-dressing during the fruit enlargement stage by spreading the fertilizer along a ditch. The melon fruits were harvested on 24 June 2016. Referring to the regular output of melon for the first two years at the trial site, the target yield was determined to be 45,000 kg/hm^2^. According to physiological properties of melon, the melon needs to uptake 157.5 kg/hm^2^ N, 77.4 kg/hm^2^ P_2_O_5_, 309.6 kg/hm^2^ K_2_O, respectively, to produce 1000 kg of melon fruit.

The coefficients were determined as follows: soil nutrient conversion factor, 2.25; vegetable land-use factor, 0.8 for general vegetables; different nutrient adjustment coefficients, 0.7 for vegetable growing season; and available soil nutrient utilization (combined with current fertilization status and related data): AN, 0.6; AP, 0.5; AK, 1.0. The nutrients available from soil and to be applied to the soil were calculated as follows:

Nutrients available from soil (kg/hm^2^) = soil available nutrients (mg/kg) × the coefficients.

Nutrient amount applied in soil = nutrient required for melons − nutrients available from soil.

N: 120.60 mg/kg × 0.8 × 0.7 × 2.25 (kg/hm^2^ × kg/mg) × 0.6 = 91.17 kg/hm^2^

P_2_O_5_: 96.65 mg/kg × 0.8 × 0.7 × 2.25 (kg/hm^2^ × kg/mg) × 0.5 = 60.89 kg/hm^2^

K_2_O: 180.01 mg/kg × 0.8 × 0.7 × 2.25 (kg/hm^2^ × kg/mg) × 1 = 226.81 kg/hm^2^

Therefore, N, P_2_O_5_, and K_2_O should be applied at the rates of 66.33, 16.51, and 82.79 kg/hm^2^ in the soil to meet the needs of melon growth, respectively. Accordingly, the application rate of carbamide, calcium superphosphate, and potassium sulfate were 360.49, 687.92, and 454.89 kg/hm^2^, respectively.

The content of soluble protein, Vc, and soluble sugar in the fruits were measured. The content of soluble protein was determined by the Coomassie blue method, Vc content was analyzed by the 2,6-dichlorophol sodium method, and soluble sugar content was measured by the anthracene colorimetric method.

Microsoft Excel 2020 was used to analyze the data, and SAS 9.2 software was used for regression analysis and mathematical model construction.

## 3. Results

### 3.1. Establishment of Mathematical Models

The contents of soluble proteins, Vc, and soluble sugars under different treatments are presented in Table 4. According to the experimental results (Table 1), a regression equation between the values of N, P_2_O_5_, and K_2_O and the content of soluble protein was established as:Y = 8.6634 − 0.0797X_1_ − 0.1733X_2_ − 0.1013X_3_ − 0.2601X_1_^2^ + 0.245X_1_X_2_ − 0.095X_1_X_3_ − 0.6842X_2_^2^ − 0.1325X_2_X_3_ − 0.3573X_3_^2^.

The lack of fit test and significance test for the function showed that the lack of fit was significant, F = 19.13 (*p* = 0.053); total regression was also significant, F = 4.66 (*p* = 0.0064), r^2^ = 0.7632. This showed that the regression equation could be used to analyze the correlation between the amount of fertilizer and the soluble protein content.

Likewise, a regression equation between the amount of N, P_2_O_5_, and K_2_O and the Vc content was established as:Y = 25.1389 − 0.697X_1_ − 0.5169X_2_ − 1.0178X_3_ − 1.2254X_1_^2^ + 0.715X_1_X_2_ + 0.495X_1_X_3_ − 2.5473X_2_^2^ − 0.935X_2_X_3_ − 0.8365X_3_^2^.

The lack of fit test and significance test for the function showed that the lack of fit was significant, F = 26.39 (*p* = 0.0061 < 0.01); total regression was also significant, F = 3.10 (*p* = 0.0319), r^2^ = 0.6820. This showed that the regression equation could be used to analyze the correlation between the amount of fertilizer and Vc content of melon.

The regression equation between the amount of N, P_2_O_5_, and K_2_O and content of soluble sugar was established as:Y = 13.2465 − 0.3197X_1_ − 0.1299X_2_ − 0.3148X_3_ − 0.2725X_1_^2^ + 0.1338X_1_X_2_ + 0.1038X_1_X_3_ − 0.8522X_2_^2^ − 0.1438X_2_X_3_ − 0.3768X_3_^2^.

The lack of fit test and significance test for the function showed that the lack of fit was significant, F = 9.13 (*p* = 0.0037 < 0.01); total regression was also significant, F = 2.85 (*p* = 0.0425), r^2^ = 0.6636. This showed that the regression equation could be used to analyze the correlation between the amount of fertilizer and soluble sugar content of melon.

### 3.2. Single-Factor Analyses

With two of the three variables in the equations set to 0, the three quadratic equations between the amount of chemical fertilizers and the content of soluble protein were established as follows:

For N fertilizer: Y = 8.6634 − 0.0797X_1_ − 0.2601X_1_^2^,

For P fertilizer: Y = 8.6634 − 0.1733X_2_ − 0.6842X_2_^2^, and

For K fertilizer: Y = 8.6634 − 0.1013X_3_ − 0.3573X_3_^2^.

According to the ranges of the three variables, three curves were drawn in Figure 1, reflecting the relationship between the content of soluble protein and the applied N, P, and K fertilizers.

The content of soluble proteins increased with the level of chemical N fertilizer in a range from −1.682 to 0.5865, attaining the maximum at a fertilization level of 0.5865. However, the soluble protein content decreased with the N level ranging from 0.5865 to 1.682. Likewise, the highest content of soluble proteins in melon was appeared when the levels of P and K fertilizers were −0.5147 and 1.1051, respectively. Therefore, it suggested that the application of K was probably the most important factor to increase the content of soluble proteins in melon; N fertilizer came second. The P fertilizer, however, affected the content of soluble proteins in melon less in this study.

With regard to the Vc content in melon, the single-factor quadratic equations were established as follows:

For N fertilizer: Y = 25.1389 − 0.697X_1_ − 1.2254X_1_^2^,

For P fertilizer: Y = 25.1389 − 0.5169X_2_ − 2.5473X_2_^2^, and

For K fertilizer: Y = 25.1389 − 1.0178X_3_ − 0.8365 X_3_^2^.

Three curves were drawn in Figure 2 to describe the changes in the amount of fertilizer application and Vc content in melon.

The Vc content in melon increased with the amount of chemical N fertilizer increasing from −1.682 to −0.2844 and reached a maximum when the N level was −0.2844. however, it decreased with the N level varying from −0.2844 to 1.682. The content of Vc was the highest when the levels of P and K fertilizers were −0.1015 and −0.5942, respectively. The curves indicated that the application of P fertilizer played the most important role in increasing Vc content in melon; K fertilizer came second; N fertilizer only ranked third. In the present study, the application of chemical N, P, and K fertilizers in a lower rate significantly increased the Vc content. After reaching the maximum level, the continuous application of K and P fertilizers would reduce the content of Vc in melon on the contrary.

With two of the three variables related to N, P, and K fertilizers set to 0, the three single-factor quadratic equations between the application rates of chemical fertilizers and content of soluble sugars were also established as follows:

For N fertilizer: Y_1_ = 13.2465 − 0.3197X_1_ − 0.2725X_1_^2^,

For P fertilizer: Y_2_ = 13.2465 − 0.1299X_2_ − 0.8522X_2_^2^, and

For K fertilizer: Y_3_ = 13.2465 − 0.3148X_3_ − 0.3768X_3_^2^.

The three curves are shown in Figure 3 to depict the effects of changes in the amount of fertilizer application on soluble sugar content. The content of soluble sugars in melon increased firstly after declining with the chemical N fertilizer application (−1.682 to 1.682), and it reached the maximum level when the N level was −0.5866. When the levels of P and K fertilizers were, respectively, −0.0762 and −0.1574, the melon exhibited the highest content of soluble sugars. The curves indicated that the application of P, K, and N fertilizers, in order, played significant roles in raising the soluble sugar content in melon.

### 3.3. Two-Factor Analyses

#### 3.3.1. Mutually Coordinated Effects of Application of N and P Fertilizers on the Quality of Melon

With the application rate of K fertilizer set to 0, a sub-model was established to reflect the changes related to the application of N and P fertilizers:Y_(1,2)_ = 8.6634 − 0.0797X_1_ − 0.1733X_2_ − 0.2600X_1_^2^ + 0.2450X_1_X_2_ − 0.6842X_2_^2^.

A 3-D map was drawn to describe the interactions between the N and P fertilizers on the content of soluble proteins in melon (Figure 4). When the levels of N and P fertilizers ranged from −1.682 to 1.682, the content of soluble proteins increased at first but then decreased notably with N and P fertilizers application increment. There was mutually coordinated effects of the combined application of N and P fertilizers on the soluble protein content. The optimal levels of N and P fertilizers were estimated as 0.2326 and 1.1683, respectively, and thus the content of soluble proteins in melon reached the maximum values.

With the application rate of K fertilizer set to 0, a sub-model was established to reflect the changes in Vc content on the basis of N and P fertilizers application:Y_(1,2)_ = 25.1389 − 0.697X_1_ − 0.5169X_2_ − 1.2254X_1_^2^ + 0.715X_1_X_2_ − 2.5473X_2_^2^.

In Figure 5, the 3-D map described the interaction between the application of N and P fertilizers and the content of Vc in melon. The Vc content increased at first but then decreased along with the increasing application rates of N and P fertilizers. Additionally, the combination of application of N and P fertilizers notably increased the Vc content with mutual coordinated effects. However, too much N and P fertilizers would reduce the Vc content in melon. The optimal levels of N and P fertilizers were estimated to be 0.2130 and 0.0671, respectively, and the Vc content reached its maximum values.

Similarly, a sub-model was established to reflect the relationship between the content of soluble sugar in melon and the amounts of N and P fertilizers:Y_(1,2)_ = 13.2465 − 0.3197X_1_ − 0.1299X_2_ − 0.2725X_1_^2^ + 0.1338X_1_X_2_ − 0.8522X_2_^2^.

A 3-D map was shown in Figure 6 to describe the variation of the soluble sugar content with different usage rates of N and P fertilizers. With the levels of N and P fertilizers ranging from −1.682 to 1.682, the content of soluble sugar in melon increased to the maximum and then decreased. The N and P fertilizers together significantly increased the content of soluble sugar in melon compared to either of them alone, indicating a mutually coordinated effect. The optimal levels of P and N fertilizers were estimated as −0.7775 and −0.2036, respectively, where the quality of melons was the best.

#### 3.3.2. Mutually Coordinated Effects of Application of P and K Fertilizers on the Quality of Melons

With the application rate of N fertilizer set to 0, a sub-model was established to reflect the changes related to the application of K and P fertilizers:Y_(2,3)_ = 8.6634 − 0.1733X_2_ − 0.1013X_3_ − 0.6842X_2_^2^ − 0.1325X_2_X_3_ − 0.3573X_3_^2^.

A 3-D map was shown in Figure 7 to describe the interactions between application of P and K fertilizers and the content of soluble proteins in melon. When the levels of K and P fertilizers ranged from −1.682 to 1.682, the content of soluble proteins in melon increased with the addition of K and P fertilizers, but then there was a decrease.

Moreover, the application of both K and P fertilizers more significantly increased the content of soluble proteins in melon, showing mutually coordinated effects. However, the continuous and simultaneous increase in K and P fertilizers reduced the content of soluble proteins. The optimal levels of K and P fertilizers were estimated as −0.1149 and −0.1204, respectively, at which the content of soluble proteins in melon reached the maximum values.

With the application rate of N fertilizer set to 0, another sub-model was established to reflect the changes related to the application of K and P fertilizers:Y_(2,3)_ = 25.1389 − 0.5169X_2_ − 1.0178X_3_ − 2.5473X_2_^2^ − 0.935X_2_X_3_ − 0.8365X_3_^2^.

In Figure 8, a 3-D map was used to describe the interactions of application of P and K fertilizers on the Vc content in melon. The Vc content tended to increase at first but then notably decrease with the application of K and P fertilizers. Similarly, the combined application of K and P fertilizers significantly increased the Vc content in melon, indicating the mutually coordinated effects. Excessive application of both K and P fertilizers would decrease the content of Vc in melon. Therefore, the optimal levels of K and P fertilizers were predicted to be 1.1672 and 0.2850, respectively, when the Vc content in melon reached its maximum values.

A sub-model was established to reflect the relationship between the content of soluble sugar and the application of K and P fertilizers, with the value of N fertilizer set to 0:Y_(2,3)_ = 13.2465 − 0.1299X_2_ − 0.3148X_3_ − 0.8522X_2_^2^ − 0.1438X_2_X_3_ − 0.3768X_3_^2^.

Then, their interactions were exhibited using the 3-D map (Figure 9). The content of soluble sugars in melon tended to increase at first but then notably decrease with the level of K and P fertilizers from −1.682 to 1.682. Mutually coordinated effects were observed with the combined application of K and P fertilizers. The continuous increase in both K and P fertilizers, however, reduced the quality of melons (e.g., soluble sugar). To obtain the best quality of melons, the optimal levels of K and P fertilizers were estimated as −0.3508 and −0.0703, respectively.

#### 3.3.3. Mutually Coordinated Effects of Application of N and K Fertilizers on the Quality of Melons

With the application level of P fertilizer set to 0, a sub-model was used to reflect the changes related to the application of N and K fertilizers:Y(1,3) = 8.6634 − 0.0797X_1_ − 0.1013X_3_ − 0.2601X_1_^2^ − 0.0950X_1_X_3_ − 0.3573X_3_^2^.

A 3-D map depicted the interactions between the application of N and K fertilizers and the content of soluble proteins in melon (Figure 10). With the increasing values of N and K fertilizers (−1.682 to 1.682), the content of soluble proteins in melon increased to the maximum but then decreased. This showed that excessive application of both N and K fertilizers was unfavorable for the accumulation of the soluble proteins. When the content of soluble proteins in melons reached the maximum values, the optimal levels of N and K fertilizers were estimated to be 0.1305 and 0.1244, respectively.

Subsequently, another sub-model was established to reflect the changes in the Vc content with the amount of P fertilizer set to 0:Y_(1,3)_ = 25.1389 − 0.697X_1_ − 1.0178X_3_ − 1.2254X_1_^2^ + 0.495X_1_X_3_ − 0.8365X_3_^2^.

In Figure 11, the 3-D map analyzed the interactions between the application of N and K fertilizers and the content of Vc in melon. The variation of the content of Vc in melon were close to a parabola going upwards and then downwards. The continuous increase in both N and K fertilizers, however, reduced the Vc content. When N and K fertilizers were applied at the same time, the Vc content significantly increased with a mutual coordinated effect. The optimal levels of N and K fertilizers were estimated as 0.0824 and 0.3125, respectively, according to the maximum Vc content in melons (Figure 11).

With P fertilizer level set to 0, the sub-model was established to reflect the changes in the soluble sugar content along with the application of N and K fertilizers:Y_(1,3)_ = 13.2465 − 0.1299X_2_ − 0.3148X_3_ − 0.8522X_2_^2^ − 0.1438X_2_X_3_ − 0.3768X_3_^2^.

A 3-D map exhibited the interactions between the application of N and K fertilizers and the content of soluble sugar (Figure 12). When the level of K fertilizers was from −1.682 to 1.682, the content of soluble sugar increased at first but then decreased significantly with K level increasing. Similar phenomenon was also noticed under the N fertilizer application condition. Mutually coordinated effects were observed with the combined application of N and K fertilizers. Excessive application of both N and K fertilizers, however, reduced the content of soluble sugar. The optimal levels of N and K fertilizers were estimated to be −0.7246 and −0.4845, respectively, according to when the quality of melons was the best.

### 3.4. Optimizing the Model through Simulation

Based on the regression model results obtained by the field experiment, a total of 125 plans were simulated to obtain the optimal scheme. When the ratio of N, P, and K levels was 0:−1:−1, the content of soluble proteins in melon was the highest (8.73 g/kg). The Vc content in melon was the highest (25.32 g/kg) when the ratio of N, P, and K levels was 0:0:−1. Furthermore, when the ratio of N, P and K levels was 1:0:−1, the content of soluble sugar in melon was the highest (13.34%) (Table 5).

## 4. Discussion

Fertilizer, the source of nutrition for vegetable crops, has an important influence on the agronomic characters and economic benefits derived from the crops. Reasonable fertilization can allow vegetable crops to achieve the goal of high yield and high quality [15]. The nutritional quality of vegetables and fruits is mainly determined by vitamins, proteins, amino acids, carbohydrates, and sugars, which are positively correlated with the nutritional value of vegetables and fruits. Many studies have shown that different N, P, and K fertilizer formulas have different effects on the quality of vegetables and fruits, which can be significantly improved through the use of a certain proportion [16,17,18].

In the experimental condition, the relationship between the combined fertilizer application and the soluble protein content was in accordance with the quadratic orthogonal regressive rotation design considering the three factors: Y = 8.6634 − 0.07974X_1_ − 0.1733X_2_ − 0.1013X_3_ − 0.2601X_1_^2^ + 0.245X_1_X_2_ − 0.095X_1_X_3_ − 0.6842X_2_^2^ − 0.1325X_2_X_3_ − 0.3573X_3_^2^. Therefore, the obtained regression model could be used for correlation analysis between the combined fertilization of N, P, and K, and soluble protein content in melon. Similarly, in accordance with the quadratic orthogonal regressive rotation design of three factors, the relationship between the combined fertilizer application and the Vc content in melon was: Y = 25.1389 − 0.697X_1_ − 0.5169X_2_ − 1.0178X_3_ − 1.2254X_1_^2^ + 0.715X_1_X_2_ + 0.495X_1_X_3_ − 2.5473X_2_^2^ − 0.935X_2_X_3_ − 0.8365X_3_^2^; the relationship between the combined fertilizer application and the soluble sugar content in melon was: Y = 13.2465 − 0.3197X_1_ − 0.1299X_2_ − 0.3148X_3_ − 0.2725X_1_^2^ + 0.1338X_1_X_2_ + 0.1038X_1_X_3_ − 0.8522X_2_^2^ − 0.1438X_2_X_3_ − 0.3768X_3_^2^. The orthogonal rotation combination design adopted in this study optimized the fertilization formula of melon, and the regression model had a good fit and significant regression. This model can provide a theoretical basis for melon fertilization.

Studies have reported that the protein content in seed pumpkin was significantly increased under a suitable application ratio of N, P, and K [19,20]. The analyses of single and double factors suggested that the factors influencing the soluble protein content were in the order of K > N > P. When the soluble protein content was used as the best index, the N, P, and K fertilization ratios were 2.33:1:3.85, i.e., the application rates of nitrogen (N), phosphorus (P_2_O_5_), potassium (K_2_O) were 157.50, 67.58 and 260.38 kg/hm^2^, respectively, and the highest soluble protein content was 8.73 mg/g. This proportion of fertilizers was consistent with the proportion described in a previous study of melons, when they reached the highest yield [19], and is also similar to the results of a study on blackberry yield and quality [20]. In addition, based on the single- and two-factor analyses of different proportions of N, P, and K, the results showed that the N fertilizer affected the Vc content, followed by P fertilizer, while the K fertilizer had the least effect, which was similar to the results of Chen [21] on balsam pear. As for the soluble sugar content in melon, P fertilizer had the greatest influence, followed by K fertilizer, and N fertilizer had the least effect, which was in accordance with the research conclusion of Gu [22].

In the experiment, a more appropriate regression model of N, P, and K fertilization was constructed based on the contents of soluble protein, Vc, and soluble sugar, as well as the target yield. Meanwhile, the single-factor and interaction effects of different fertilizers were analyzed, in order to provide a theoretical basis for melon production. As melon production is not only affected by fertilizer application, but also by the combined effects of local climate, soil fertility, irrigation, and many other factors. These factors should to be considered in future studies, in order to provide better guidance for the agricultural production practice.

## 5. Conclusions

In our experiment, through simulation and optimization, we concluded that, when the N, P, and K fertilization ratio was 2.33:1:3.85—that is, the application rates of nitrogen (N), phosphorus (P_2_O_5_), and potassium (K_2_O) were 157.50, 67.58, and 260.38 kg/hm^2^, respectively—the soluble protein content in melon could be maximized (at 8.73 g/kg). When the N, P, and K fertilization ratio was 2.03:1:3.36—that is, the application rates of nitrogen (N), phosphorus (P_2_O_5_), and potassium (K_2_O) were 57.50, 77.40, and 260.38 kg/hm^2^, respectively—the Vc content could reach the highest value (25.32 g/kg). Finally, when the N, P, and K fertilization ratio was 1.53:1:3.36—that is, the application rates of nitrogen (N), phosphorus (P_2_O_5_), and potassium (K_2_O) were 118.07, 77.40, and 260.38 kg/hm^2^, respectively—the content of soluble sugar of melon was maximized (at 13.34%).

## Figures and Tables

**Figure 1 ijerph-20-00283-f001:**
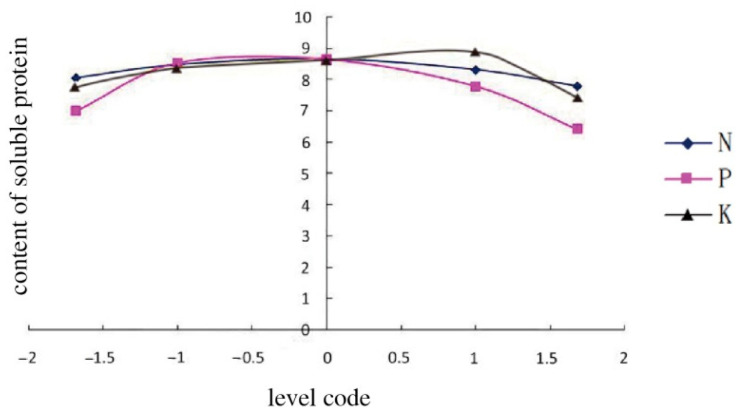
The effect of the application amount of N, P, and K on the soluble protein content in melon.

**Figure 2 ijerph-20-00283-f002:**
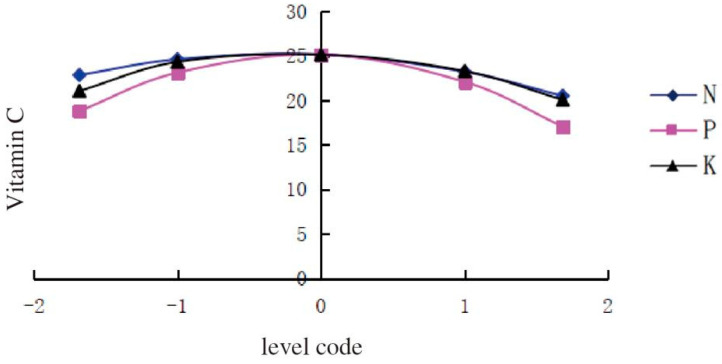
The effects of N, P, and K application rates on the Vc content in melon.

**Figure 3 ijerph-20-00283-f003:**
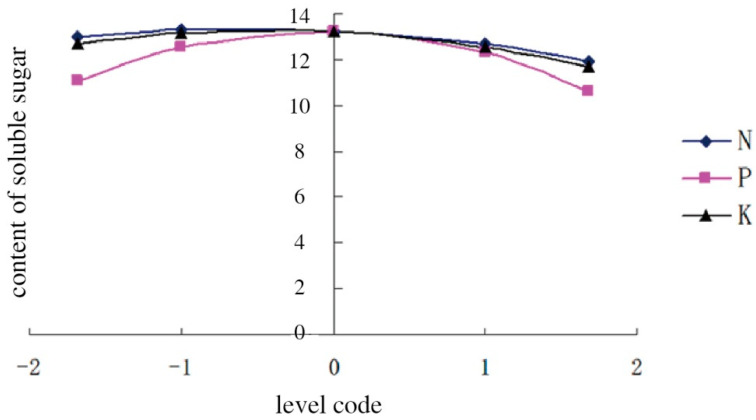
The effect of the application of N, P, and K fertilizers on the content of soluble sugars in melon.

**Figure 4 ijerph-20-00283-f004:**
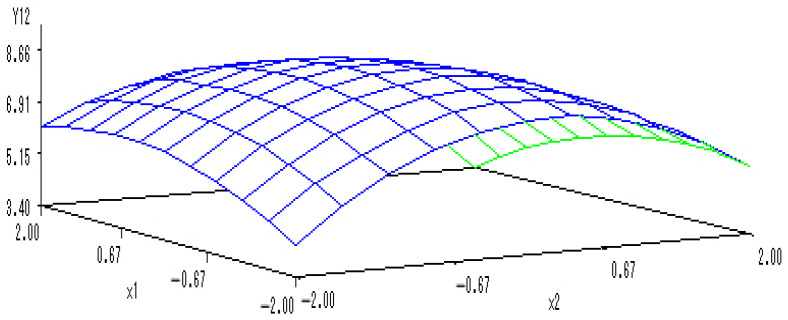
The 3-D map of mutual effect of N and P fertilizers on content of soluble protein content in melon.

**Figure 5 ijerph-20-00283-f005:**
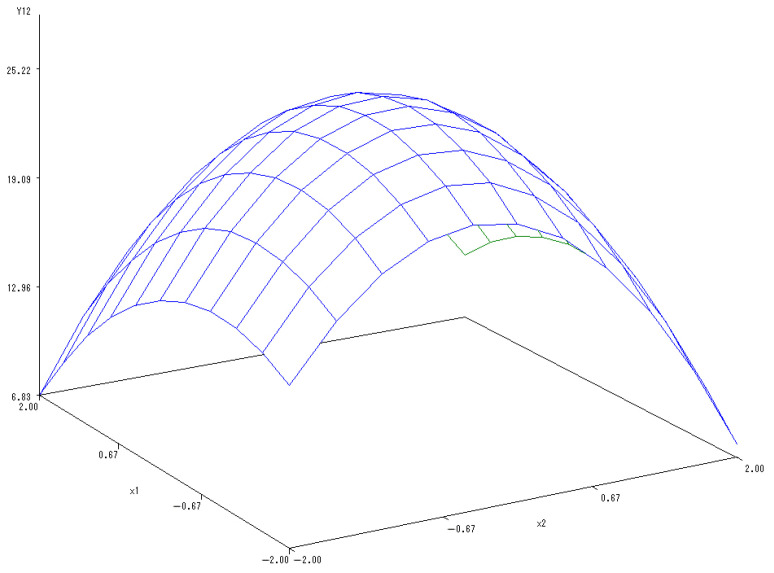
The 3-D map of mutual effect of N and P fertilizers application on the content of Vc in melon.

**Figure 6 ijerph-20-00283-f006:**
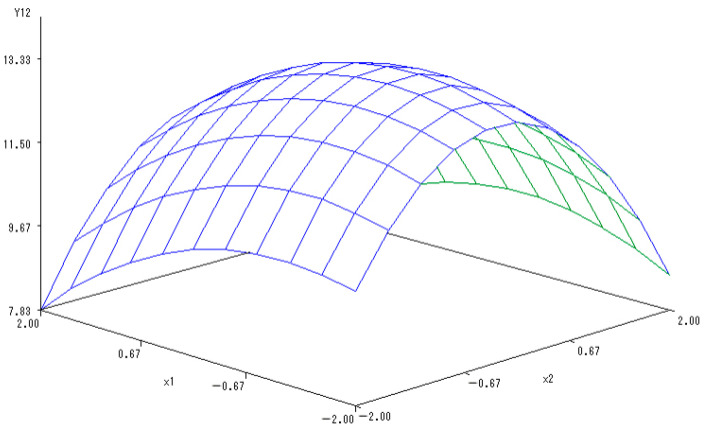
The 3-D map of mutual effect of N and P fertilizers application on the content of soluble sugar in melon.

**Figure 7 ijerph-20-00283-f007:**
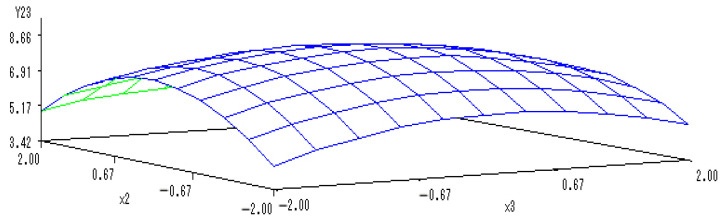
The 3-D map of mutual effect of P and K fertilizers on the content of soluble proteins in melon.

**Figure 8 ijerph-20-00283-f008:**
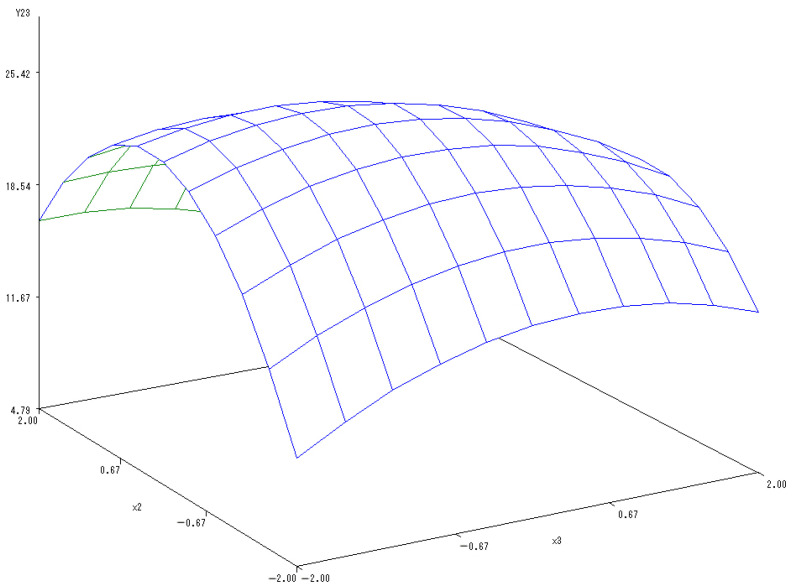
The 3-D map of mutual effect of P and K fertilizers application on Vc content in melon.

**Figure 9 ijerph-20-00283-f009:**
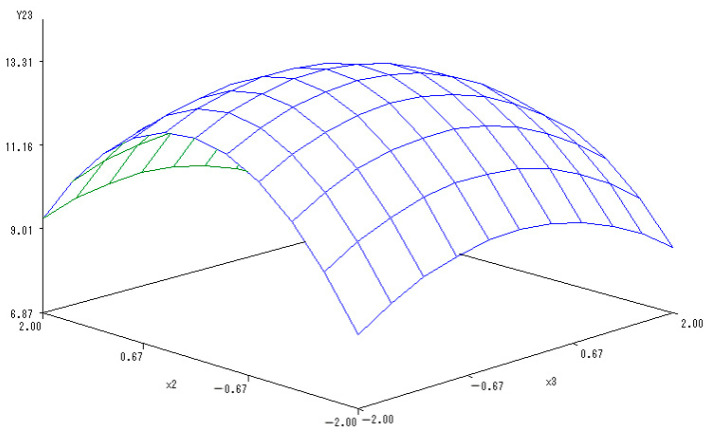
The 3-D map of mutual effect of P and K fertilizers application on the content of soluble sugar in melon.

**Figure 10 ijerph-20-00283-f010:**
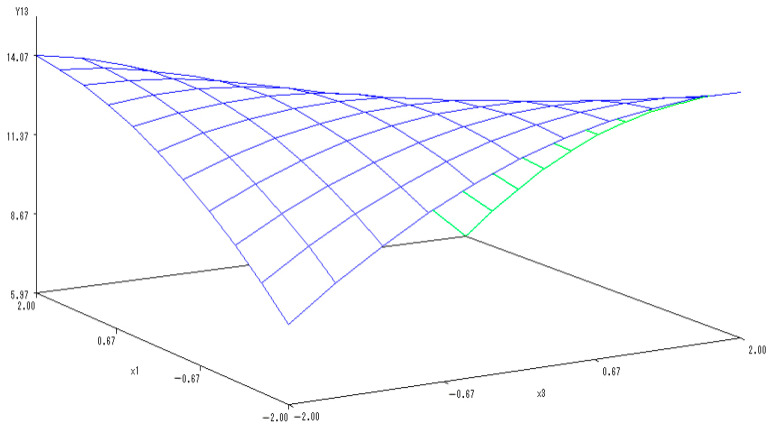
The 3-D map of mutual effect of N and K fertilizers application on the soluble protein content in melon.

**Figure 11 ijerph-20-00283-f011:**
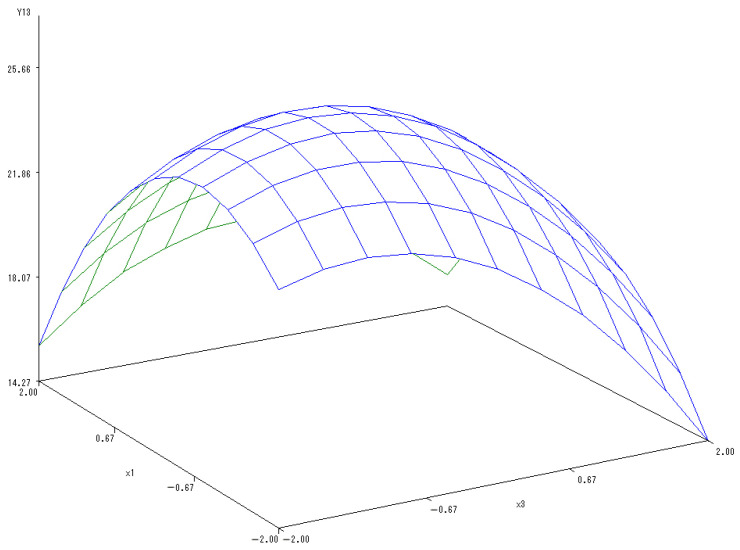
The 3-D map of mutual effect of N and K fertilizers application on Vc content in melon.

**Figure 12 ijerph-20-00283-f012:**
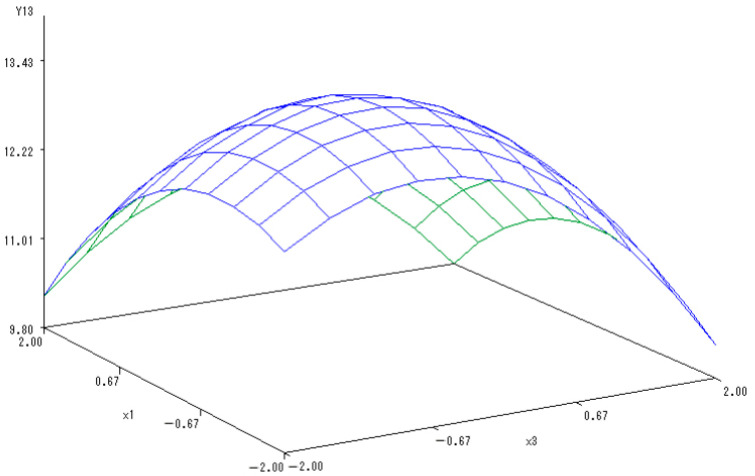
The 3-D map of mutual effect of N and K fertilizers application on content of soluble sugar in melon.

**Table 1 ijerph-20-00283-t001:** Content and utilization of fertilizer nutrients.

Fertilizer	Available Nutrient Content	Utilization Rate
Carbamide	N ≥ 46%	40%
Calcium superphosphate	P_2_O_5_ ≥ 12%	25%
Potassium sulfate	K_2_O ≥ 50%	15%

**Table 2 ijerph-20-00283-t002:** Fertilizer application rate and linear coding data.

Fertilizer	−1.682	−1	0	1	1.682
Carbamide	0	146.17	360.49	574.81	720.98
Calcium superphosphate	0	278.93	687.92	1096.91	1375.84
Potassium sulfate	0	184.44	454.89	725.34	909.78

**Table 3 ijerph-20-00283-t003:** Trial structure matrix and different treatments in the experiment.

Treatment	X_1_		X_2_		X_3_	
	Linear Code	Application Amount (kg/hm^2^)	Linear Code	Application Amount (kg/hm^2^)	Linear Code	Application Amount (kg/hm^2^)
1	1	574.81	1	1096.91	1	725.34
2	1	574.81	1	1096.91	−1	184.44
3	1	574.81	−1	278.93	1	725.34
4	1	574.81	−1	278.93	−1	184.44
5	−1	146.17	1	1096.91	1	725.34
6	−1	146.17	1	1096.91	−1	184.44
7	−1	146.17	−1	278.93	1	725.34
8	−1	146.17	−1	278.93	−1	184.44
9	1.682	720.98	0	687.92	0	454.89
10	−1.682	0	0	687.92	0	454.89
11	0	360.49	1.682	1375.84	0	454.89
12	0	360.49	−1.682	0	0	454.89
13	0	360.49	0	687.92	1.682	909.78
14	0	360.49	0	687.92	−1.682	0
15	0	360.49	0	687.92	0	454.89
16	0	360.49	0	687.92	0	454.89
17	0	360.49	0	687.92	0	454.89
18	0	360.49	0	687.92	0	454.89
19	0	360.49	0	687.92	0	454.89
20	0	360.49	0	687.92	0	454.89
21	0	360.49	0	687.92	0	454.89
22	0	360.49	0	687.92	0	454.89
23	0	360.49	0	687.92	0	454.89

Note: X_1_, X_2_, and X_3_ refer to carbamide, calcium superphosphate, and potassium sulfate, respectively. The same as below. The linear codes were determined by the quadratic regression orthogonal rotation combination design principle in multivariate statistics.

**Table 4 ijerph-20-00283-t004:** The contents of soluble protein, Vc, and soluble sugar under the different treatments.

Treatment	X_1_	X_2_	X_3_	Soluble Protein Content (mg/g)	Vc (g/kg)	Soluble Sugar Content (%)
1	1	1	1	7.52	20.24	11.8018
2	1	1	−1	7.55	21.56	11.6872
3	1	−1	1	7.49	22.00	11.8152
4	1	−1	−1	7.43	19.36	11.8330
5	−1	1	1	7.39	20.68	11.9893
6	−1	1	−1	7.48	23.76	12.9908
7	−1	−1	1	8.78	25.08	13.2274
8	−1	−1	−1	7.90	24.64	12.9566
9	1.682	0	0	7.60	19.80	11.6113
10	−1.682	0	0	7.32	18.92	11.8122
11	0	1.682	0	6.05	14.96	9.9595
12	0	−1.682	0	6.47	16.28	10.1946
13	0	0	1.682	6.53	16.72	10.3316
14	0	0	−1.682	7.84	24.2	12.5027
15	0	0	0	8.76	25.08	13.3628
16	0	0	0	8.65	25.52	13.3345
17	0	0	0	8.80	23.76	12.3003
18	0	0	0	8.91	25.08	13.3389
19	0	0	0	8.82	24.64	13.3182
20	0	0	0	8.29	25.52	13.9029
21	0	0	0	8.51	25.08	13.2051
22	0	0	0	8.75	26.40	13.3375
23	0	0	0	8.64	25.96	13.3762

**Table 5 ijerph-20-00283-t005:** Analysis of fertilizer levels leading to the best quality in melons.

Fertilizer	Factor	Combination with the Highest Soluble Protein Content		Combination with the Highest Vc Content		Combination with the Highest Soluble Sugar Content	
		Coding Quantity	Actual Fertilizer Dosage (kg/hm^2^)	Coding Quantity	Actual Fertilizer Dosage (kg/hm^2^)	Coding Quantity	Actual Fertilizer Dosage (kg/hm^2^)
Carbamide	X_1_	0	360.49	0	360.49	−1	146.17
Calcium superphosphate	X_2_	−1	278.93	0	687.92	0	687.92
Potassium sulfate	X_3_	−1	184.44	−1	184.44	−1	184.44

## Data Availability

The datasets used and analyzed during the current study are available from the corresponding author on reasonable request.

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
