# Peer review of "Using Mathematical Models to Study the Influences of Different Ratios of Chemical Nitrogen, Phosphorus, and Potassium on the Content of Soluble Protein, Vitamin C, and Soluble Sugar in Melon"

_ijerph, 2022, doi:10.3390/ijerph20010283_

Round 1

Reviewer 1 Report

Comments to the Author

‘Using mathematical models to study the influences of different ratios of chemical nitrogen, phosphorus, and potassium on the content of soluble protein, Vitamin C, and soluble sugar in melon’ is remarkable research in this field.

Here is a general summary of my observations and suggestions. Detailed notes and questions can be found in follow.

The topic and research questions posed in this manuscript are certainly interesting, and the results would benefit growers, as well as improve our understanding of the nutritional management in Melon. However, the manuscript needs to be edited for grammar, punctuation mistakes, inconsistencies, and general writing style.

Page 8- line 237: level increasing … dot.

Page 11- line 307: of Nand K fertilizers on the …space N and K

I can recommend acceptance with minor review.

There follow some more specific questions and comments on the text.

(Suggest to describe the answer of first Question in your discussion section.)

·         In materials and methods, you did not determine the content of mineral nutrients in the soil. Could the different amounts of micronutrient elements in plots cause change in your results?  For instance, S content of soil could affect Protein content in fruit. How do you describe this effect? What about Fe, Ze, Mn etc.

·        Could you mention the general climate condition of your experiment and Shanghai (briefly)?

·        Do you have any other effective parameters that you did not expect in your experiments and could affect the data?

·        Regarding your harvest, how many fruit per plants did you check for sampling? Did you get one fruit per plant or more?

Reviewer 2 Report

THE PAPER SEEMS GOOD BUT THE FOLLOWING CHANGES MAY BE INCORPORATED:

1. THE INTERACTION EFFECT  OF THE FACTORSIN THE MATHEMATICAL MODEL MUST BE PRESENTED AND PRESENT THE ANOVA TABLE AS WELL TO SEE THE SIGNIFICANACE OF THE VARIABLES.

2.PRESENT THE R SAQUARED AND LACK FOR FIT FOR THE MODELS.

3. PRSENT THE CRITEIA FOR OPTIMIZATION AND PUT THE VALIDATION OF RESULTS AT OPTIAL CONDITIONS.

4. REPORT THE SIGNIFICANCE OF THE FIT AND REPLACATION BY DUNCANS POST HOC TEST

5. EXPERIMENTAL PLAN NEED CLEA DEPICTION.

Reviewer 3 Report

Below are my comments for the manuscript.

Abstract:

None

Introduction:

All of the requirements for good nutrient management weren’t addressed: Right fertilizer source, right rate, right time, place

Line 45: Please add the word “in” after especially. Also, remove the word “is”

Materials and Methods:

Line 72: Please explain the 3 factors.

Line 73: Shouldn’t there be 207 test plots? 3 factors x 23 treatments x 3 replications

Line 76: What rate were the melons planted? Were the excess plants thinned (removed) to get a final stand of 10 plants per plot?

Please add more information about how the fertilizers were applied.

Table 2: Please add more information to table 2 to explain the numbers.

Table 3: Please explain how the linear codes were determined.

Line 87: Please explain base fertilizer.

Line 91: Was the experiment repeated? Fertilizer uptake is highly influenced by the environment. Was any environmental data used in the modeling?

Results:

Putting the results in terms of kg/hm2 would be more useful for producers or researchers performing agricultural research.

Sub-heading 3.1: Please change modals to models

Line 162 and throughout the text: Please change “singe-factor” to “single-factor”

Line 179: Please change “till reaches the” to “until”

Line 193: Please remove the sentence “But it decreases…”

Line 237: Please add a “.” After increasing

Discussion:

None

Conclusions:

None
